# Overseas background executives and enterprise ESG performance -The moderating effect of the executive pay gap

**Baoping Liu[1]\***, **Haohao Wei[2]**, **Jing Liu[3]**

1 School of Business, Henan University, Kaifeng, Henan, People's Republic of China, 2 School of Economics, Henan University, Kaifeng, Henan, People's Republic of China, 3 Digital Communication Research Center, Konkuk University, Seoul, Korea

\* liubaoping2005@163.com

## Abstract

Enhancing corporate ESG (Environmental, Social, and Governance) performance is a critical issue garnering widespread attention across various sectors. This research examines the impact of executives with overseas backgrounds on corporate ESG performance, utilizing a two-way fixed-effects model with data from Chinese A-share listed companies in Shanghai and Shenzhen spanning from 2008 to 2022. The findings indicate that: (i) executives with overseas backgrounds positively influence corporate ESG performance, with this effect amplifying as the proportion of such executives within the executive team increases; (ii)both internal and external pay gaps significantly positively moderate the relationship between overseas background executives and corporate ESG performance; (iii) the influence of overseas executives on ESG performance is more pronounced in state-owned enterprises, large-scale firms, firms in the eastern region, and those in high-pollution industries. This study expands the understanding of factors affecting corporate ESG performance, offers empirical insights for listed companies aiming to enhance their ESG performance, and provides valuable implications for executive team construction, executive compensation strategies, government talent policies, and the pursuit of high-quality economic development.

## 1. Introduction

With the deepening implementation of the talent-driven development strategy, China's economy has transitioned from a phase of rapid growth to a stage of high-quality development. Under the new era of economic globalization and continuous enhancement of comprehensive national strength, China has witnessed a remarkable "reverse flow of talent" in recent years, with over a million high-level overseas professionals choosing to return for domestic development [1]. A significant proportion of these returnees have assumed core managerial positions in local enterprises [2].

**Data availability statement:** All relevant data are within the manuscript and its Supporting Information files.

**Funding:** This work is supported by the Major Program Grants for Applied Research from the Education Department of Henan Province (2023-YYZD-04), China. The funders had no role in study design, data collection and analysis, decision to publish, or preparation of the manuscript.

**Competing interests:** The authors have declared that no competing interests exist.

These versatile talents with international perspectives, global resources, and transnational networks [3] have not only introduced advanced governance philosophies and technical expertise from abroad [4], but also played a pivotal bridging role in driving national industrial upgrading and stimulating corporate innovation [5]. They constitute indispensable strategic resources for China's economic transformation and social reform [6]. In practical terms, while enterprises require substantial high-level talent to ensure sustainable development, the imperative for sustainability simultaneously demands that corporations establish ESG systems and strengthen ESG management. This dual requirement conversely compels businesses to employ more specialized technical experts and senior executives. Given the complex interplay between corporate leadership and sustainable development [7], coupled with growing attention from investors and financiers to environmental, social, and governance (ESG) standards [8], it becomes critically important to thoroughly investigate whether and how executives with overseas backgrounds influence corporate ESG performance.

Existing research has shown that executives with overseas backgrounds can enhance corporate R&D investment and the number of patent applications [9], effectively improve corporate operations and boost corporate performance [10], strengthen corporate risk-taking and enterprise performance [11], and further promote green innovation [3]. Moreover, academic studies on the factors that influence corporate ESG performance can be categorized into macro and micro levels. At the macro level, factors such as national economic development levels [12], regional cultural differences [13], changes in party and government leadership in cities [14], government green financial policies [15], sustainable public procurement [16], and market competition shocks [17] can influence corporate ESG performance. At the micro level, corporate ESG performance is also affected by factors such as firm size [18], performance feedback [19], online attention [20], managerial myopia [21], and board size and diversity [22]. Overall, current domestic and international research lacks detailed studies on the relationship between executives with overseas backgrounds and corporate ESG performance. A preliminary literature search reveals that only Liu et al. [23] have indicated that Chinese CEOs' overseas experience improves corporate ESG performance. However, the CEO is only one member of the corporate executive team, which differs from the focus of this study. According to Upper Echelons Theory, the background and experiences of top executives may influence corporate decision-making by shaping their perceptions and interpretations of external environments [24]. Therefore, there remains a research gap regarding the impact of overseas background executives on corporate ESG performance, and this gap motivates the first key research question of our study:

RQ1. Can executives with overseas backgrounds influence corporate ESG performance?

Additionally, the issue of pay gaps has attracted widespread societal attention, which underscores the potential moderating role of executive pay gaps in the relationship between overseas background executives and corporate ESG performance. Cohen et al. [25] note that incorporating ESG metrics into executive compensation contracts has become an internationally prevalent practice. Meng et al. [26] believe that linking

executive pay to ESG performance can enhance ESG outcomes, while ESG practices, in turn, may increase executive compensation. Fu et al. [27] further reveal that a greater number of overseas background executives correlates with larger internal pay disparities within firms. According to tournament theory, compensation gaps among executives participating in promotion tournaments can serve as effective incentives. Widening pay disparities may motivate executives to work harder, reduce agency costs, and improve corporate performance [28]. Building on this analysis, this paper categorizes executive pay gaps into internal and external disparities to address our second research question:

RQ2. Do executive pay gaps moderate the impact of overseas background executives on corporate ESG performance?

To address the issues mentioned above, this study utilizes panel data from China's Shanghai and Shenzhen A-share listed companies spanning 2008–2022 as the research sample. We employ a two-way fixed effects model to analyze whether and how overseas background executives influence corporate ESG performance, as well as how internal and external executive pay gaps facilitate this relationship. The findings reveal that executives with overseas backgrounds significantly enhance corporate ESG performance. Both internal and external executive pay gaps exhibit positive moderating effects on the relationship between overseas background executives and ESG performance. Further investigation into the heterogeneity of core relationships demonstrates that the impact of overseas background executives on ESG performance varies depending on the ownership type, size, geographic location, and industry sector of the enterprises.

The contributions of this study include the following: First, this paper integrates overseas background executives and corporate ESG performance into a unified analytical framework to investigate how executive characteristics enhance ESG performance. This approach broadens research on factors influencing corporate ESG performance and enriches scholarly achievements in the ESG field. Second, the study focuses on the moderating role of executive pay gaps in the relationship between overseas background executives and ESG performance. It provides theoretical guidance for designing corporate compensation systems and incentive mechanisms, while offering actionable insights for firms to refine executive compensation contracts, improve operational efficiency, and strengthen governance practices. Third, by examining heterogeneity in ownership structure, firm size, geographic location, and industry attributes, this research reveals differentiated effects of overseas background executives on corporate ESG performance. These findings provide empirical evidence to advance high-quality development in enterprises.

The remaining sections of this study are structured as follows. Section 2 presents the theoretical framework and hypotheses. Section 3 describes the data, variables, and empirical models. Section 4 reports the empirical analysis and results, followed by robustness checks in Section 5. Section 6 discusses the moderating effects and heterogeneity analysis. Finally, Section 7 concludes the study.

## 2. Theoretical analysis and research hypotheses

### 2.1 Overseas background executives and corporate ESG performance

The concept of ESG can be traced back to the "Who Cares Wins" report released by the United Nations in 2004, which evaluates corporate green and sustainable development performance across three dimensions: environmental, social responsibility, and corporate governance [29]. Research indicates that capital markets are increasingly prioritizing corporate green practices and sustainable development [30], with investors showing a greater inclination toward companies with strong ESG performance [31]. ESG scores have emerged as a comprehensive metric for assessing corporate sustainability [32]. Against this backdrop, senior executives—as primary decision-makers and implementers of corporate strategies—are driven to adopt green development principles, comply with environmental regulations, and advance ESG initiatives [33].

According to the Upper Echelons Theory, the educational and professional experiences of executive team members reshape their values, thereby influencing their strategic choices [34]. Executives with overseas backgrounds—those who have studied or worked abroad—are influenced by the values and cognitive frameworks of foreign countries and regions,

 

which further shape their strategic decision-making behaviors. Giannetti et al. [10] found that overseas background executives apply advanced technologies and practices learned abroad to domestic firms, thereby enhancing corporate performance. In China, executives with overseas backgrounds primarily come from Western developed countries. They integrate knowledge from ESG frameworks in Europe and the U.S., and possess robust theoretical and practical foundations in ESG, which enables them to formulate and implement ESG strategies, ultimately improving corporate ESG performance [35]. Therefore, overseas background executives are influenced by the ESG system and culture of developed countries during the critical period of growth. Their risk preferences, decision-making inclinations, and final choices will enhance the effectiveness and scientific rigor of ESG strategies. Additionally, given the limitations of individual knowledge structures and cognitive dimensions, the proportion of overseas background executives within the leadership team also plays a critical role in corporate ESG performance. Based on this analysis, the following hypotheses are proposed:

**H1a.** Executives with overseas backgrounds significantly promote corporate ESG performance.

**H1b.** A higher proportion of overseas background executives correlates with better corporate ESG performance.

## 2.2 Overseas background management, executive pay gap, and corporate ESG performance

Research indicates that incentivizing executives is one of the critical priorities for enterprises [36]. Sufficient incentives are conducive to mitigating the short-sighted behavior of corporate managers and enhancing their willingness to participate in risky projects [37]. Moreover, as an incentive mechanism, the executive pay gap influences the daily financial decisions and strategic behaviors of executives [38]. According to tournament theory, competition among executive team members resembles a tournament, where winners gain benefits such as higher compensation, reputation, and industry influence. The pay gap represents additional rewards for outstanding performance, with larger gaps attracting greater attention from executives [39]. Expanding this gap can also amplify the tournament incentive effect within industries, motivate executives to work harder, improve organizational efficiency and financial decision-making, and ultimately enhance innovation investment and corporate performance [40]. This study examines the moderating role of executive pay gaps—both internal (within the firm) and external (relative to industry peers)—in the relationship between overseas background executives and corporate ESG performance.

From the perspective of internal executive pay gaps, three key aspects can be analyzed. First, executive pay gaps serve as an incentive mechanism that enhances corporate governance quality and aligns executives' interests with those of shareholders, thereby motivating management to improve corporate performance and value [41]. Second, compensation gaps foster a competitive atmosphere within organizations, stimulating rivalry among executives at different levels. This drives their desire for promotions and salary increases, maintains high work enthusiasm, and facilitates strategic transformations [42], including proactive fulfillment of corporate responsibilities in environmental, social, and governance (ESG) domains. Third, compensation gaps can attract top-tier talent, and infuse new vitality into organizational development [27]. The addition of high-quality professionals will bring advanced technologies, management expertise, and ESG-oriented sustainability principles. These individuals prioritize long-term corporate interests, elevate long-term value, reduce speculative investments and short-termism, and actively enhance corporate ESG performance.

From the perspective of external executive pay gaps, the analysis can be divided into two aspects. First, people always like to compare with others to get positive emotional satisfaction, which is the nature of human beings. When executives discover that their compensation exceeds the industry average among peers outside their organization, they will feel that their abilities are recognized, thus generating a sense of superiority [43]. This superiority can generate a positive incentive effect, satisfy psychological needs and enhance motivation and work engagement. Conversely, if executives find their compensation is lower than the external industry average, they may attribute this to social inequity, leading to dissatisfaction, reduced motivation, and lower productivity. Second, from the standpoint of the managerial labor market, when the market is mature and efficient, a widening external compensation gap strengthens the motivation for lower-paid executives to seek alternative incentive compensation. By demonstrating performance and building reputations through effort, they enhance

their bargaining power for "promotions and salary increases" within the labor market [44]. Meanwhile, higher-paid executives strive to maintain their elevated compensation and established reputations by working harder to improve corporate performance and gain shareholder recognition [45]. Therefore, when overseas background executives perceive external pay gaps, they become more dedicated to their roles, and make strategic decisions aligned with long-term corporate development goals. This drives efforts to enhance long-term interests and value, and ultimately improve corporate ESG performance.

Based on the above analysis, this study proposes the following hypotheses:

**H2a.** The internal executive pay gap exerts a positive moderating effect on the relationship between overseas background executives and ESG performance.

**H2b.** The external executive pay gap exerts a positive moderating effect on the relationship between overseas background executives and ESG performance.

The theoretical model is depicted in Fig 1.

## 3. Research design

### 3.1 Sample and data source

This study selects data from China's Shanghai and Shenzhen A-share listed companies spanning 2008–2022 as the research sample. The initial sample was filtered and processed as follows: (1) We excluded companies in the financial and insurance industries. (2) We removed all observations labeled as ST or *ST. (3) We omitted samples with severe missing data for key variables. The final dataset comprises 12,949 firm-year observations. Corporate ESG performance data were sourced from the Bloomberg database, while financial and executive characteristic data were obtained from the CSMAR database. All continuous variables in the sample were winsorized at the 1% and 99% levels to mitigate the impact of outliers on regression results.

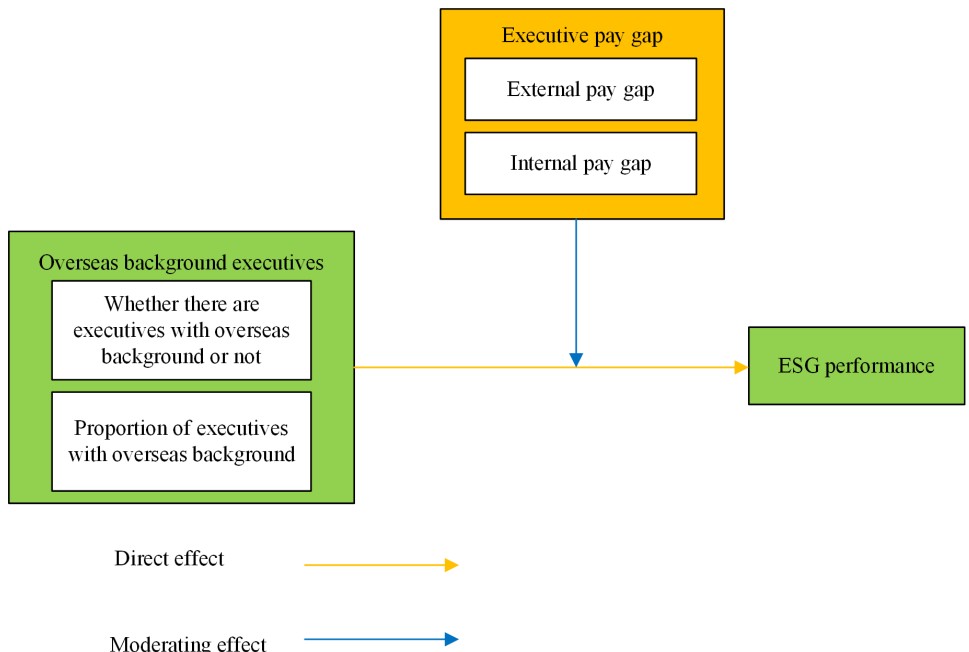

**Fig 1. Theoretical model.**

## 3.2 Measure of main variables

**3.2.1 Dependent variable: Corporate ESG performance (*ESG*).** Current academic research, both domestically and internationally, measures corporate ESG performance primarily through constructing evaluation frameworks for itemized and composite scoring or by directly adopting ratings published by third-party authoritative agencies. Drawing on the methodology of Qing et al. [8], this study utilizes the ESG disclosure scores provided by Bloomberg L.P. to gauge corporate ESG performance. Specifically, Bloomberg's quantitative ESG system generates numerical data, offering not only overall ESG scores but also dimension-specific scores for environmental, social responsibility, and corporate governance performance. This approach ensures objectivity, professionalism, and independence in the assessment.

**3.2.2 Independent variables: Overseas background executives (*Oversea1* and *Oversea2*).** Existing studies define senior executives as individuals directly involved in corporate management and decision-making, such as chief executive officers (CEOs), chief financial officers (CFOs), general managers, deputy general managers, board chairs, vice chairs, board secretaries, and other managers disclosed in annual reports [46]. In this study, corporate executives are defined as individuals disclosed in listed companies' annual reports, including directors, supervisors, board secretaries, general managers, core technical personnel, and department heads. Given the significant differences in economic systems, education levels, and historical-cultural contexts between Mainland China and Hong Kong, Macau, and Taiwan regions, the scope of "overseas backgrounds" in this paper is expanded to include these regions.

Drawing on the methodology of Xie & Wang [47], the explanatory variables are operationalized in two distinct measurement approaches: ① Presence of Overseas-Experienced Executives (Oversea1): A binary variable set to 1 if the firm has executives with overseas backgrounds in a given year, and zero otherwise. ② Proportion of Overseas-Experienced Executives (Oversea2): A continuous variable measured as the percentage of executives with overseas backgrounds relative to the total executive team in a given year, enhancing the robustness of the analysis.

**3.2.3 Moderating variable: Executive pay gap (*Paygap1* and *Paygap2*).** We measure executive compensation gaps from both internal and external perspectives. First, following the approach of Niu et al. [48], we adjust the units of executive compensation to balance the scale of metrics. Second, drawing on Faleye et al. [49], executives are categorized into core executives (top three highest-paid) and non-core executives (others). Finally, adopting the methodology of Gu & Zhu [50], the internal executive pay gap (*Paygap1*) is defined as the absolute gap between the average compensation of core executives and non-core executives. The external executive pay gap (*Paygap2*) is measured as the absolute gap between the average compensation of core executives and the industry-wide average compensation.

**3.2.4 Control variables.** Drawing on the methodologies of Ren et al. [19] and Fan et al. [21], we select the following control variables: firm-level financial characteristics including firm size (*Size*), listing age (*ListAge*), leverage ratio (*Lev*), return on equity (*ROE*), and growth potential (*Growth*). Corporate governance characteristics are controlled using board size (Board), proportion of independent directors (*Indep*), CEO duality (*Dual*), and the shareholding ratio of the largest shareholder (*Top1*). Additionally, year and industry dummy variables are included to account for year-specific and industry-specific effects. Detailed definitions of all variables are provided in **Table 1**.

## 3.3 Model setting

We construct Model (1) to examine the relationship between executives with overseas backgrounds (*Oversea*) and corporate ESG performance (*ESG*):

$$ESG_{it} = \alpha_0 + \alpha_1 Oversea_{it} + \alpha_2 Controls_{it} + \pi_t + \rho_c + \varepsilon_{it} \tag{1}$$

Among them, the explanatory variable is *ESG*, which indicates the ESG performance score of the enterprise, and *Oversea* is the independent variable, including whether the enterprise has an overseas background executive (*Oversea1*) and the proportion of the enterprise having an overseas background executive (*Oversea2*). *Controls* represent control variables,

**Table 1. Definitions of key variables.**

| Type | Name | Symbol | Variable definition |
|---|---|---|---|
| *Dependent variable* | The firm's ESG performance | *ESG* | The firm's ESG score published by Bloomberg |
| *Independent variables* | Overseas background executives | *Oversea1* | Whether there are executives with overseas background in the firm in the year, with a value of 1, otherwise 0 |
| | | *Oversea2* | Overseas executives/all executives in the firm in that year |
| *Moderating variables* | The executive internal pay gap | *Paygap1* | abs (average pay of core executives - average pay of non-core executives) |
| | The executive external pay gap | *Paygap2* | abs (average pay of core executives - average pay of industry executives) |
| *Control Variables* | Firm Size | *Size* | The natural logarithm of the total business assets |
| | Years listed | *ListAge* | The total number of listed years of the firm |
| | Gearing ratio | *Lev* | Total liabilities at the end of the period/ Total assets at the end of the period |
| | Return on net assets | *Roe* | Net profit from the main business of the year/net assets at the end of the period |
| | Growth | *Growth* | (Current operating income - previous operating income)/previous operating income |
| | Size of Board of Directors | *Board* | Natural logarithm of the number of directors |
| | Proportion of independent directors | *Indep* | The proportion of independent directors on the board of directors |
| | Dual role of the chairman and general manager | *Dual* | If the chairman and general manager are the same person, the value is 1; otherwise, the value is 0 |
| | The share ratio of the largest shareholder | *Top1* | The proportion of the shares of the largest shareholder |
| | Year | *Year* | Annual virtual variable |
| | Industry | *Industry* | Industry dummy variables |

π and ρ represent time-fixed effects and industry-fixed effects respectively, and ε represents random error terms. Furthermore, in order to test the moderating effect of the executive pay gap (*Paygap*) on the relationship between overseas executives and the ESG performance score, we introduce the interaction term between the above variables and *Oversea* based on the Model (1). The concrete model is as follows:

$$ESG_{it} = \beta_0 + \beta_1 Oversea_{it} + \beta_2 M_{it} + \beta_3 Oversea_{it} \times M_{it} + \beta_4 Controls_{it} + \pi_t + \rho_c + \varepsilon_{it} \qquad (2)$$

Here, M stands for executive pay gap (*Paygap*), which is further divided into executive internal pay gap (*Paygap1*) and executive external pay gap (*Paygap2*). Other terms retain the same definitions as in Model (1).

## 4. Empirical analysis and results

### 4.1 Descriptive statistics

Table 2 presents the descriptive statistics of the main variables. The dependent variable, ESG score, has a mean value of 28.141 and a standard deviation of 9.524, indicating substantial variation in ESG performance across firms. The minimum and maximum values are 10.703 and 57.479, respectively. Compared to the full score of 100 in Bloomberg's ESG disclosure ratings, these results suggest significant room for improvement in ESG performance among the sample firms.

For the independent variables, the mean values of Oversea1 and Oversea2 are 0.262 and 0.062, respectively. This implies that 26.2% of firms employ executives with overseas backgrounds. In comparison the average proportion of overseas executives relative to the total executive team is approximately 6.2%, reflecting the relatively low prevalence of such executives in China's listed companies.

**Table 2. Descriptive statistics of variables.**

| Variables | Obs | Mean | SD | Min | Max |
|---|---|---|---|---|---|
| ESG | 12949 | 28.141 | 9.524 | 10.703 | 57.479 |
| Oversea1 | 12949 | 0.262 | 0.439 | 0.000 | 1.000 |
| Oversea2 | 12949 | 0.062 | 0.133 | 0.000 | 1.000 |
| Paygap1 | 12416 | 0.826 | 0.791 | 0.044 | 3.987 |
| Paygap2 | 12416 | 0.778 | 0.923 | 0.004 | 4.411 |
| Size | 12949 | 23.111 | 1.310 | 19.317 | 26.452 |
| ListAge | 12949 | 2.455 | 0.670 | 0.000 | 3.401 |
| Lev | 12949 | 0.483 | 0.196 | 0.030 | 0.908 |
| Roe | 12949 | 0.091 | 0.127 | -0.926 | 0.437 |
| Growth | 12949 | 0.186 | 0.419 | -0.658 | 4.024 |
| Board | 12949 | 2.183 | 0.202 | 1.609 | 2.708 |
| Indep | 12949 | 37.434 | 5.523 | 25.000 | 60.000 |
| Dual | 12949 | 0.193 | 0.395 | 0.000 | 1.000 |
| Top1 | 12949 | 37.943 | 15.980 | 8.020 | 75.843 |

Regarding the moderating variables, the internal pay gap (Paygap1) and external pay gap (Paygap2) have mean values of 0.826 and 0.778, with standard deviations of 0.791 and 0.923, respectively. These figures highlight notable disparities in compensation gaps across firms.

For other control variables, all exhibit significant heterogeneity in their distributions, underscoring the diverse characteristics of the sampled firms. This variation confirms the appropriateness of including these variables to account for firm-specific financial and governance factors.

## 4.2 Correlation analysis

The results of the Pearson correlation test for the main variables are shown in **Table 3**. The presence of overseas executives (Oversea1) and the proportion of overseas executives (Oversea2) are significantly and positively correlated with the ESG scores of enterprises at the 1% level. This preliminary finding suggests that, without accounting for other factors, firms with overseas-experienced executives and a higher proportion of such executives exhibit better ESG performance, providing initial support for hypotheses H1a and H1b. The low correlation coefficient among the variables indicates that there is no ]multicollinearity between the variables in this paper. In addition, the VIF test in **Table 4** implicates that the mean-variance inflation factor is 1.62 for all variables, while the maximum value for variables is 2.67, with no value exceeding 5, further illustrating the absence of multicollinearity between explanatory variables.

## 4.3 Regression analysis

This study conducts regression analysis by incorporating the explanatory variables Oversea1 and Oversea2 into Model (1) using Stata 18.0. **Table 5** presents the OLS regression results. Without any control variables, Columns (1) and (4) show that the regression coefficients for the presence of overseas-experienced executives (Oversea1) and their proportion (Oversea2) are 1.933 and 5.465, respectively, both statistically significant at the 1% level. This preliminary finding suggests that overseas background executives significantly enhance corporate ESG performance, and a higher proportion of such executives further strengthens ESG outcomes.

After controlling for firm-level financial characteristics (Columns 2 and 5), the positive relationship between overseas-experienced executives and ESG performance remains statistically significant at the 1% level, though the coefficients decrease slightly. Further inclusion of corporate governance controls (Columns 3 and 6) reveals that

**Table 3. Correlation analysis.**

| Variables | ESG | Oversea1 | Oversea2 | Size | ListAge | Lev |
|---|---|---|---|---|---|---|
| ESG | 1 | | | | | |
| Oversea1 | 0.182*** | 1 | | | | |
| Oversea2 | 0.171*** | 0.781*** | 1 | | | |
| Size | 0.472*** | 0.088*** | 0.055*** | 1 | | |
| ListAge | 0.183*** | -0.095*** | -0.094*** | 0.249*** | 1 | |
| Lev | 0.030*** | -0.028*** | -0.054*** | 0.474*** | 0.230*** | 1 |
| Roe | 0.011 | 0.010 | 0.022** | 0.034*** | -0.164*** | -0.208*** |
| Growth | -0.010 | 0.038*** | 0.044*** | 0.010 | -0.106*** | 0.019** |
| Board | -0.028*** | -0.025*** | -0.060*** | 0.176*** | 0.053*** | 0.112*** |
| Indep | 0.089*** | 0.027*** | 0.035*** | 0.084*** | -0.008 | 0.013 |
| Dual | 0.035*** | 0.089*** | 0.072*** | -0.108*** | -0.177*** | -0.104*** |
| Top1 | -0.013 | -0.093*** | -0.066*** | 0.218*** | -0.092*** | 0.079*** |
| Variables | Roe | Growth | Board | Indep | Dual | Top1 |
| Roe | 1 | | | | | |
| Growth | 0.286*** | 1 | | | | |
| Board | 0 | -0.038*** | 1 | | | |
| Indep | 0.002 | 0.008 | -0.428*** | 1 | | |
| Dual | 0.059*** | 0.051*** | -0.179*** | 0.081*** | 1 | |
| Top1 | 0.111*** | 0.010 | 0.039*** | 0.076*** | -0.096*** | 1 |

**Table 4. Variance inflation factor (VIF).**

| Variables | VIF | 1/VIF |
|---|---|---|
| Ovesrea1 | 2.67 | 0.374 |
| Oversea2 | 2.66 | 0.376 |
| Size | 2.03 | 0.492 |
| ListAge | 1.38 | 0.725 |
| Lev | 1.51 | 0.664 |
| Roe | 1.26 | 0.795 |
| Growth | 1.12 | 0.895 |
| Board | 1.36 | 0.733 |
| Indep | 1.27 | 0.789 |
| Dual | 1.11 | 0.900 |
| Top1 | 1.45 | 0.684 |
| Mean VIF | 1.62 | |

overseas-experienced executives exert a statistically significant positive driving effect on ESG performance, with coefficients of 1.207 and 3.921, respectively, both passing the 1% significance test.

Thus, hypotheses H1a and H1b are validated. These results indicate that overseas-experienced executives leverage their advanced knowledge, technical expertise, and cutting-edge management philosophies to improve corporate performance in social responsibility, environmental protection, and governance practices.

These findings broadly align with those of Liu et al. [23], who reported a positive effect of CEO foreign experience on corporate ESG performance. Furthermore, they share similarities with Chen et al. [3], who identified a positive association

**Table 5. Impact of overseas background executives on ESG performance.**

| Variables | (1) ESG | (2) ESG | (3) ESG | (4) ESG | (5) ESG | (6) ESG |
|---|---|---|---|---|---|---|
| Oversea1 | 1.933*** | 1.215*** | 1.207*** | | | |
| | (0.148) | (0.130) | (0.131) | | | |
| Oversea2 | | | | 5.465*** | 3.904*** | 3.921*** |
| | | | | (0.540) | (0.483) | (0.487) |
| Size | | 2.367*** | 2.297*** | | 2.391*** | 2.320*** |
| | | (0.058) | (0.060) | | (0.059) | (0.061) |
| ListAge | | -0.579*** | -0.607*** | | -0.585*** | -0.614*** |
| | | (0.093) | (0.095) | | (0.093) | (0.094) |
| Lev | | -2.516*** | -2.498*** | | -2.481*** | -2.469*** |
| | | (0.350) | (0.349) | | (0.352) | (0.351) |
| Roe | | 2.796*** | 2.874*** | | 2.725*** | 2.813*** |
| | | (0.421) | (0.422) | | (0.420) | (0.421) |
| Growth | | -0.336*** | -0.317*** | | -0.338*** | -0.318*** |
| | | (0.119) | (0.119) | | (0.120) | (0.119) |
| Board | | | 1.361*** | | | 1.476*** |
| | | | (0.309) | | | (0.308) |
| Indep | | | 0.070*** | | | 0.071*** |
| | | | (0.011) | | | (0.011) |
| Dual | | | -0.142 | | | -0.116 |
| | | | (0.135) | | | (0.135) |
| Top1 | | | -0.003 | | | -0.004 |
| | | | (0.004) | | | (0.004) |
| _cons | 27.636*** | -24.436*** | -28.223*** | 27.805*** | -24.908*** | -28.904*** |
| | (0.059) | (1.262) | (1.419) | (0.058) | (1.270) | (1.425) |
| Year FE | Yes | Yes | Yes | Yes | Yes | Yes |
| Industry FE | Yes | Yes | Yes | Yes | Yes | Yes |
| adj. $R^2$ | 0.578 | 0.644 | 0.645 | 0.576 | 0.644 | 0.645 |
| F | 170.994 | 345.641 | 213.077 | 102.263 | 342.742 | 211.486 |
| N | 12949 | 12949 | 12949 | 12949 | 12949 | 12949 |

Note: In parentheses are robust standard errors, * * *, * * and * are significance levels of 1%, 5% and 10%, respectively.

between executives' overseas background and corporate green innovation. In essence, as senior managers, overseas background executives act as both strategists and implementers—balancing short-term organizational performance with the need to prioritize long-term development. Compared to prior studies, this research advances the literature by employing dual measurement approaches (presence and proportion of overseas-experienced executives) to provide a granular exploration of the relationship between such executives and ESG performance.

## 5. Robustness test

### 5.1 Propensity score matching

Executives with overseas backgrounds who are exposed to political, economic, and cultural influences from developed countries may be more inclined to choose companies with good ESG performance. Therefore, there may be sample

selection bias in the research process. In order to further guarantee the reliability of the benchmark regression results, the propensity score matching method (PSM) was used to control the sample selection bias.

Firstly, the virtual variable of overseas executives (*Oversea1*) was used as the dependent variable, the enterprises with overseas executives as the treatment group, and the enterprises without overseas executives as the control group. The matching was conducted based on the nearest neighbor with a ratio of 1:1, allowing replacements. The results of the balance test in **Table 6** show that after covariates matched the treatment and control groups, the bias of each covariate between the treatment and control groups was significantly reduced. Except for the board size bias of 4.1%, the absolute values of the other covariates were all less than 3.0%, which indicated that the matching effect was good.

Table 7 shows the regression results for the Propensity Score Matching re-estimation Model (1). As can be seen from Table 7, 2573 matched samples were obtained, and a total of 5146 samples were obtained from the control group. After matching, the regression coefficients of *Oversea1* and *Oversea2* were significantly positive at the 1% level. Column (2) shows that the ESG of enterprises with overseas executives performs better, and the influence coefficient of selecting overseas executives on the ESG of enterprises is 1.097, which passes the significance test at the level of 1%; column (4) shows that the ESG of enterprises with overseas executives performs better, and with the proportion of overseas executives (*Oversea2*) rising, the impact of overseas executives on ESG performance improvement becomes more obvious. Therefore, the conclusion of this study is still valid after the bias of sample selection is controlled by the propensity score matching method.

## 5.2 Replace the independent variable

First, drawing on the methodology of Wang Kai et al. [51], we calculate the number of overseas background executives (*Oversea3*) and substitute this variable for the original explanatory variables in the baseline regression to test the robustness of the findings. As shown in Column (1) of **Table 8**, the number of overseas background executives significantly enhances corporate ESG performance, with the coefficient passing the 1% significance test. This confirms that the research conclusions remain robust even after substituting the explanatory variables, demonstrating that the results are not affected by the measurement approach of the variables.

**Table 6. Results of balance test.**

| Covariates | UMatched/Matched | Treated | Control | %bias | T value | P value |
|---|---|---|---|---|---|---|
| *Size* | U | 23.305 | 23.042 | 19.9 | 10.1 | 0 |
| | M | 23.305 | 23.284 | 1.6 | 0.65 | 0.517 |
| *ListAge* | U | 2.348 | 2.493 | -21.3 | -10.89 | 0 |
| | M | 2.348 | 2.361 | -1.9 | -0.73 | 0.466 |
| *Lev* | U | 0.473 | 0.486 | -6.4 | -3.2 | 0.001 |
| | M | 0.473 | 0.475 | -0.8 | -0.33 | 0.739 |
| *Roe* | U | 0.093 | 0.090 | 2.3 | 1.18 | 0.238 |
| | M | 0.093 | 0.092 | 1.1 | 0.46 | 0.646 |
| *Growth* | U | 0.213 | 0.177 | 8.6 | 4.3 | 0 |
| | M | 0.213 | 0.213 | -0.1 | -0.06 | 0.956 |
| *Board* | U | 2.175 | 2.186 | -5.7 | -2.85 | 0.004 |
| | M | 2.175 | 2.166 | 4.1 | 1.71 | 0.087 |
| *Indep* | U | 37.685 | 37.345 | 6.1 | 3.07 | 0.002 |
| | M | 37.685 | 37.758 | -1.3 | -0.53 | 0.595 |
| *Dual* | U | 0.252 | 0.172 | 19.7 | 10.17 | 0 |
| | M | 0.252 | 0.264 | -2.8 | -1.06 | 0.291 |
| *Top1* | U | 35.441 | 38.828 | -21.2 | -10.65 | 0 |
| | M | 35.441 | 35.069 | 2.3 | 0.97 | 0.332 |

**Table 7. The influence of overseas executives on ESG performance by PSM.**

| Variables | (1) ESG | (2) ESG | (3) ESG | (4) ESG |
|---|---|---|---|---|
| Oversea1 | 1.308*** | 1.097*** | | |
| | (0.196) | (0.173) | | |
| Oversea2 | | | 3.483*** | 3.562*** |
| | | | (0.675) | (0.612) |
| Size | | 2.842*** | | 2.849*** |
| | | (0.102) | | (0.103) |
| ListAge | | -0.858*** | | -0.848*** |
| | | (0.151) | | (0.151) |
| Lev | | -2.207*** | | -2.078*** |
| | | (0.595) | | (0.597) |
| Roe | | 3.766*** | | 3.782*** |
| | | (0.704) | | (0.702) |
| Growth | | -0.304 | | -0.316 |
| | | (0.195) | | (0.195) |
| Board | | 0.964* | | 1.209** |
| | | (0.561) | | (0.557) |
| Indep | | 0.058*** | | 0.060*** |
| | | (0.018) | | (0.018) |
| Dual | | 0.015 | | 0.028 |
| | | (0.210) | | (0.210) |
| Top1 | | 0.003 | | 0.002 |
| | | (0.006) | | (0.006) |
| _cons | 28.818*** | -38.679*** | 29.068*** | -39.370*** |
| | (0.123) | (2.425) | (0.114) | (2.435) |
| Year FE | Yes | Yes | Yes | Yes |
| Industry FE | Yes | Yes | Yes | Yes |
| adj. $R^2$ | 0.533 | 0.628 | 0.532 | 0.628 |
| F | 44.487 | 113.928 | 26.627 | 112.998 |
| N | 5146 | 5146 | 5146 | 5146 |

Note: In parentheses are robust standard errors, $* * *$, $* *$ and $*$ are significance levels of 1%, 5% and 10%, respectively.

Second, following Liu et al. [23], this study refines the analysis by precisely targeting overseas-experienced executives in specific leadership roles, such as CEOs and board chairs. We construct the virtual variables of the overseas background CEO (*Oversea4*) and overseas background chairman (*Oversea5*). The regression results are shown in columns (2) and (3) of **Table 8**. It can be seen that both overseas background CEOs and board chairs exert a statistically significant positive impact on corporate ESG performance, with coefficients significant at the 1% level. This further validates the robustness of the baseline conclusions for hypotheses H1a and H1b, confirming that the findings are not contingent on the operationalization of the explanatory variables.

## 5.3 Replace the dependent variable

In addition to disclosing the overall ESG score (ESG), Bloomberg provides separate detailed scores for environmental (E), social (S), and governance (G) dimensions. Therefore, we substitute the composite ESG score in the baseline

 

**Table 8. The estimated result after replacing the independent variable.**

| Variables | (1) ESG | (2) ESG | (3) ESG |
|---|---|---|---|
| Oversea3 | 0.835*** | | |
| | (0.080) | | |
| Oversea4 | | 0.800*** | |
| | | (0.199) | |
| Oversea5 | | | 1.151*** |
| | | | (0.215) |
| Size | 2.269*** | 2.357*** | 2.365*** |
| | (0.060) | (0.061) | (0.061) |
| ListAge | -0.578*** | -0.681*** | -0.667*** |
| | (0.094) | (0.095) | (0.094) |
| Lev | -2.432*** | -2.490*** | -2.496*** |
| | (0.349) | (0.350) | (0.350) |
| Roe | 2.813*** | 2.808*** | 2.743*** |
| | (0.420) | (0.420) | (0.420) |
| Growth | -0.321*** | -0.295** | -0.287** |
| | (0.119) | (0.119) | (0.119) |
| Board | 1.341*** | 1.449*** | 1.427*** |
| | (0.307) | (0.310) | (0.310) |
| Indep | 0.071*** | 0.071*** | 0.071*** |
| | (0.011) | (0.011) | (0.011) |
| Dual | -0.155 | -0.107 | -0.111 |
| | (0.135) | (0.135) | (0.135) |
| Top1 | -0.002 | -0.005 | -0.005 |
| | (0.004) | (0.004) | (0.004) |
| _cons | -27.703*** | -29.341*** | -29.539*** |
| | (1.413) | (1.433) | (1.433) |
| Year FE | Yes | Yes | Yes |
| Industry FE | Yes | Yes | Yes |
| adj. $R^2$ | 0.647 | 0.643 | 0.644 |
| F | 216.113 | 208.484 | 207.640 |
| N | 12949 | 12949 | 12949 |

Note: In parentheses are robust standard errors, * * *, * * and * are significance levels of 1%, 5% and 10%, respectively.

regression with the dimension-specific scores (E, S, G) and re-estimate the model. The results, as shown in Table 9, indicate that both the presence of overseas background executives (*Oversea1*) and their proportion (*Oversea2*) exhibit statistically significant positive effects on corporate performance across all three ESG dimensions: environmental, social, and governance.

Furthermore, the enhancing effects of overseas background executives vary markedly across dimensions. As we can see in Columns (1) and (4) of Table 9, the strongest impact is observed in environmental performance, where the coefficients for *Oversea1* and *Oversea2* are 2.031 and 7.540, respectively, both significant at the 1% level. This highlights the differential influence of overseas-experienced executives on distinct ESG pillars, with environmental initiatives benefiting most from their expertise.

**Table 9. The estimated result after replacing the dependent variable.**

| Variables | (1) E | (2) S | (3) G | (4) E | (5) S | (6) G |
|---|---|---|---|---|---|---|
| Oversea1 | 2.031*** | 0.728*** | 0.939*** | | | |
| | (0.225) | (0.144) | (0.177) | | | |
| Oversea2 | | | | 7.540*** | 2.604*** | 1.571*** |
| | | | | (0.856) | (0.543) | (0.572) |
| Size | 3.080*** | 1.768*** | 1.873*** | 3.107*** | 1.779*** | 1.908*** |
| | (0.102) | (0.064) | (0.090) | (0.103) | (0.064) | (0.090) |
| ListAge | -1.037*** | -1.075*** | -0.570*** | -1.025*** | -1.073*** | -0.613*** |
| | (0.168) | (0.110) | (0.132) | (0.167) | (0.110) | (0.132) |
| Lev | -2.905*** | -3.417*** | -2.141*** | -2.840*** | -3.396*** | -2.142*** |
| | (0.599) | (0.389) | (0.496) | (0.602) | (0.390) | (0.496) |
| Roe | 4.927*** | 2.903*** | -0.162 | 4.833*** | 2.869*** | -0.223 |
| | (0.735) | (0.451) | (0.599) | (0.734) | (0.451) | (0.599) |
| Growth | -0.381** | -0.501*** | -0.140 | -0.390** | -0.504*** | -0.130 |
| | (0.192) | (0.122) | (0.188) | (0.193) | (0.122) | (0.188) |
| Board | 0.724 | 1.627*** | 1.703*** | 0.936* | 1.701*** | 1.763*** |
| | (0.527) | (0.349) | (0.446) | (0.522) | (0.346) | (0.446) |
| Indep | 0.052*** | 0.056*** | 0.109*** | 0.053*** | 0.056*** | 0.110*** |
| | (0.018) | (0.012) | (0.016) | (0.018) | (0.012) | (0.016) |
| Dual | -0.132 | -0.357** | -0.004 | -0.094 | -0.343** | 0.026 |
| | (0.235) | (0.154) | (0.186) | (0.235) | (0.154) | (0.186) |
| Top1 | 0.001 | -0.011*** | -0.002 | 0.000 | -0.012*** | -0.003 |
| | (0.006) | (0.004) | (0.005) | (0.006) | (0.004) | (0.005) |
| _cons | -63.044*** | -28.980*** | 14.136*** | -64.117*** | -29.373*** | 13.492*** |
| | (2.430) | (1.563) | (2.141) | (2.436) | (1.560) | (2.147) |
| Year FE | Yes | Yes | Yes | Yes | Yes | Yes |
| Industry FE | Yes | Yes | Yes | Yes | Yes | Yes |
| adj. $R^2$ | 0.385 | 0.343 | 0.686 | 0.387 | 0.344 | 0.686 |
| F | 134.464 | 128.884 | 64.165 | 135.543 | 128.495 | 62.564 |
| N | 12949 | 12949 | 12949 | 12949 | 12949 | 12949 |

Note: In parentheses are robust standard errors, * * *, * * and * are significance levels of 1%, 5% and 10%, respectively.

### 5.4 Consider the impact of the year

The empirical analysis in this study utilizes a sample spanning 2008–2022, an extended timeframe during which results may be influenced by policy shifts or socio-economic events relevant to the research focus. Notable examples include the 2008 global financial crisis, the 2015 Chinese stock market crash, and the 2020 COVID-19 pandemic—all of which imposed new demands on corporate sustainable development and significantly impacted ESG performance, potentially introducing bias to baseline regression conclusions. However, given that 2008 marks the starting year of our dataset and pre-2008 data is unavailable for comparison, the 2008 global financial crisis is excluded from the event study analysis. Instead, we focus on the 2015 financial turbulence and the 2020 COVID-19 pandemic outbreak, which disrupted global socio-economic systems.

We partition the sample into two groups based on the event year to examine whether overseas background executives' ESG-enhancing effects persist before and after the shocks. Results are presented in **Table 10**. Columns (1) – (4) demonstrate that before and after the 2015 Chinese financial crisis, overseas background executives significantly improved

**Table 10. Consider the impact of the year.**

| Variables | (1) By 2015 | (2) After 2015 | (3) By 2015 | (4) After 2015 | (5) By 2020 | (6) After 2020 | (7) By 2020 | (8) After 2020 |
|---|---|---|---|---|---|---|---|---|
| Oversea1 | 1.306*** | 0.992*** | | | 1.189*** | 1.178*** | | |
| | (0.165) | (0.203) | | | (0.140) | (0.313) | | |
| Oversea2 | | | 4.472*** | 2.203*** | | | 3.847*** | 3.521*** |
| | | | (0.600) | (0.707) | | | (0.523) | (1.036) |
| Size | 2.782*** | 1.662*** | 2.797*** | 1.695*** | 1.965*** | 3.582*** | 1.991*** | 3.602*** |
| | (0.081) | (0.088) | (0.081) | (0.089) | (0.065) | (0.147) | (0.065) | (0.148) |
| ListAge | -0.576*** | -0.696*** | -0.564*** | -0.740*** | -0.509*** | -0.882*** | -0.519*** | -0.893*** |
| | (0.131) | (0.125) | (0.131) | (0.126) | (0.097) | (0.253) | (0.097) | (0.252) |
| Lev | -2.253*** | -2.636*** | -2.177*** | -2.650*** | -2.477*** | -1.451 | -2.474*** | -1.323 |
| | (0.482) | (0.478) | (0.485) | (0.479) | (0.355) | (1.006) | (0.356) | (1.015) |
| Roe | 3.032*** | -0.140 | 2.942*** | -0.102 | 1.652*** | 2.836*** | 1.627*** | 2.744*** |
| | (0.525) | (0.705) | (0.524) | (0.708) | (0.466) | (0.988) | (0.466) | (0.989) |
| Growth | -0.270* | -0.322* | -0.275* | -0.320* | -0.483*** | 0.646 | -0.480*** | 0.625 |
| | (0.147) | (0.190) | (0.147) | (0.191) | (0.116) | (0.412) | (0.116) | (0.414) |
| Board | 1.532*** | 1.018** | 1.661*** | 1.094*** | 1.595*** | 0.441 | 1.703*** | 0.571 |
| | (0.443) | (0.398) | (0.439) | (0.400) | (0.313) | (0.902) | (0.313) | (0.893) |
| Indep | 0.065*** | 0.058*** | 0.066*** | 0.059*** | 0.068*** | 0.056* | 0.068*** | 0.057* |
| | (0.015) | (0.014) | (0.015) | (0.014) | (0.011) | (0.030) | (0.011) | (0.030) |
| Dual | -0.135 | -0.009 | -0.122 | 0.043 | -0.152 | -0.095 | -0.120 | -0.088 |
| | (0.175) | (0.193) | (0.175) | (0.193) | (0.138) | (0.346) | (0.138) | (0.346) |
| Top1 | 0.013** | -0.019*** | 0.012** | -0.020*** | -0.006 | 0.025** | -0.007* | 0.024** |
| | (0.005) | (0.005) | (0.005) | (0.005) | (0.004) | (0.010) | (0.004) | (0.010) |
| _cons | -36.160*** | -18.361*** | -36.817*** | -19.033*** | -23.187*** | -48.545*** | -23.885*** | -49.242*** |
| | (1.979) | (1.868) | (1.979) | (1.889) | (1.487) | (3.685) | (1.498) | (3.669) |
| Year FE | Yes | Yes | Yes | Yes | Yes | Yes | Yes | Yes |
| Industry FE | Yes | Yes | Yes | Yes | Yes | Yes | Yes | Yes |
| adj. $R^2$ | 0.452 | 0.214 | 0.453 | 0.211 | 0.583 | 0.393 | 0.582 | 0.393 |
| F | 179.748 | 50.217 | 180.199 | 48.959 | 132.183 | 95.182 | 130.401 | 95.056 |
| N | 7964 | 4983 | 7964 | 4983 | 10282 | 2663 | 10282 | 2663 |

Note: In parentheses are robust standard errors, ***, ** and * are significance levels of 1%, 5% and 10%, respectively.

corporate ESG performance. The coefficients for *Oversea1* are 1.306 (pre-crisis) and 0.992 (post-crisis), while those for *Oversea2* are 4.472 and 2.203, respectively—all statistically significant at the 1% level. Columns (5) – (8) reveal that the positive effects of overseas background executives on ESG performance persisted through the 2020 COVID-19 pandemic, with *Oversea1* coefficients of 1.189 (pre-pandemic) and 1.178 (post-pandemic), and *Oversea2* coefficients of 3.847 and 3.521, respectively, all passing the 1% significance test. These findings confirm the robustness of hypotheses H1a and H1b, as the baseline conclusions remain stable across distinct exogenous shocks.

## 6. Further analysis

### 6.1 Moderating effect analysis

Prior research indicates that executives' overseas experiences are closely associated with corporate compensation disparities [27], and such pay gaps can further influence ESG-related behaviors [42]. Building on these insights, we

hypothesize that executive pay gaps may act as a moderating mechanism in the relationship between overseas background executives and ESG performance.

Table 11 presents the moderating effects of executive pay gaps. According to column (1), the interaction term between overseas background executives and the internal pay gap (*Oversea1\*Paygap1*) yields a regression coefficient of 1.229, significant at the 1% level. This suggests that internal pay gaps significantly amplify the positive impact of overseas background executives on ESG performance. According to column (3), the interaction term between the proportion of overseas background executives and the internal pay gap (*Oversea2\*Paygap1*) produces a coefficient of 1.791, also significant at the 1% level. This reinforces the enhancing role of internal pay gaps, thereby validating hypothesis H2a. These results confirm that the internal executive pay gap serves as a catalytic moderator, strengthening the ESG-enhancing effects of overseas-experienced executives.

Similarly, in Column (2), the interaction term between overseas background executives and the external pay gap (*Oversea2\*Paygap1*) yields a regression coefficient of 1.132, significant at the 1% level. In Column (4), the interaction term between the proportion of overseas background executives and the external pay gap (*Oversea2\*Paygap2*) produces a coefficient of 1.769, also significant at the 1% level. These results collectively indicate that external executive pay gaps positively moderate the relationship between overseas background executives and corporate ESG performance, thereby validating hypothesis H2b.

## 6.2 Heterogeneity analysis

The impact of overseas background executives on corporate development may vary depending on factors such as ownership type [52] and geographical location [53]. To address these variations, we incorporate heterogeneity in ownership structure, firm size, region, and industry to account for potential divergences in their effects.

**6.2.1 Analysis of ownership heterogeneity.** The impact of overseas background executives on corporate ESG performance may differ between state-owned enterprises (SOEs) and non-SOEs due to inherent variations in their governance structures and strategic priorities. To examine this heterogeneity, we construct a dummy variable (SOE) where SOE = 1 for state-owned firms and SOE = 0 for non-state-owned firms. Regression results from Columns (1) and (2) of Table 12 reveal that the interaction terms *Oversea1\*SOE* and *Oversea2\*SOE* yield coefficients of 1.890 and 6.759, respectively, both statistically significant at the 1% level. This indicates that overseas-experienced executives enhance ESG performance more effectively in SOEs compared to non-SOEs.

A plausible explanation is that State-owned enterprises (SOEs) possess significant scale advantages and economic strength, granting them greater market competitiveness and bargaining power. Moreover, they bear heightened economic, political, and social responsibilities compared to non-SOEs. Consequently, the institutional environment of SOEs provides stronger incentives and structural support for overseas background executives to actively fulfill social responsibilities, engage in environmental governance, and implement decisions that enhance corporate ESG performance ratings.

**6.2.2 Analysis of scale heterogeneity.** Differences in firm size lead to variations in capital accumulation, green innovation capabilities, and sustainability management systems, which may result in heterogeneous effects of overseas-experienced executives on ESG performance. To examine this, we categorize firms into two groups based on the annual median of firm size (Scale): firms above the median are assigned Scale = 1, and those below Scale = 0. The re-estimated results of Model (1) are shown in Columns (3) and (4) of Table 12. It can be seen that the interaction terms *Oversea1\*Scale* and *Oversea2\*Scale* yield coefficients of 2.416 and 6.044, respectively, both statistically significant at the 1% level. This indicates that overseas-experienced executives exert a stronger ESG-enhancing effect in larger firms compared to smaller counterparts.

A plausible explanation is that larger firms typically possess greater capabilities and advantages in R&D innovation, resource integration, risk resilience, and strategic planning. These strengths enable overseas-experienced executives to leverage enhanced motivation and institutional resources to improve corporate ESG performance.

**Table 11. The estimated result of moderating effect analysis.**

| Variables | (1) ESG | (2) ESG | (3) ESG | (4) ESG |
|---|---|---|---|---|
| Oversea1 | 0.802*** | 0.774*** | | |
| | (0.130) | (0.129) | | |
| Oversea2 | | | 1.625*** | 1.595*** |
| | | | (0.484) | (0.480) |
| Paygap1 | 0.543*** | | 0.993*** | |
| | (0.112) | | (0.098) | |
| Paygap2 | | 0.469*** | | 0.875*** |
| | | (0.093) | | (0.081) |
| Oversea1*Paygap1 | 1.229*** | | | |
| | (0.169) | | | |
| Oversea1*Paygap2 | | 1.132*** | | |
| | | (0.145) | | |
| Oversea2*Paygap1 | | | 1.791*** | |
| | | | (0.422) | |
| Oversea2*Paygap2 | | | | 1.769*** |
| | | | | (0.372) |
| Size | 2.110*** | 2.067*** | 2.128*** | 2.084*** |
| | (0.064) | (0.065) | (0.065) | (0.065) |
| ListAge | -0.555*** | -0.536*** | -0.577*** | -0.552*** |
| | (0.095) | (0.095) | (0.095) | (0.095) |
| Lev | -2.343*** | -2.316*** | -2.291*** | -2.275*** |
| | (0.362) | (0.362) | (0.363) | (0.363) |
| Roe | 1.937*** | 1.828*** | 1.737*** | 1.639*** |
| | (0.466) | (0.466) | (0.465) | (0.465) |
| Growth | -0.333*** | -0.327*** | -0.322*** | -0.317*** |
| | (0.121) | (0.121) | (0.121) | (0.121) |
| Board | 1.168*** | 1.190*** | 1.232*** | 1.256*** |
| | (0.313) | (0.313) | (0.314) | (0.314) |
| Indep | 0.075*** | 0.076*** | 0.076*** | 0.077*** |
| | (0.011) | (0.011) | (0.011) | (0.011) |
| Dual | -0.112 | -0.110 | -0.082 | -0.081 |
| | (0.137) | (0.136) | (0.137) | (0.137) |
| Top1 | -0.003 | -0.002 | -0.004 | -0.003 |
| | (0.004) | (0.004) | (0.004) | (0.004) |
| _cons | -24.116*** | -23.193*** | -24.832*** | -23.873*** |
| | (1.461) | (1.470) | (1.467) | (1.476) |
| adj. $R^2$ | 0.655 | 0.656 | 0.653 | 0.653 |
| F | 184.747 | 186.752 | 177.924 | 179.904 |
| N | 12414 | 12414 | 12414 | 12414 |

Note: In parentheses are robust standard errors, * * *, * * and * are significance levels of 1%, 5% and 10%, respectively.

**Table 12. Results of ownership and scale heterogeneity analysis.**

| Variables | Heterogeneity of property rights | | Heterogeneity of scale | |
| --- | --- | --- | --- | --- |
| | (1) | (2) | (3) | (4) |
| | *ESG* | *ESG* | *ESG* | *ESG* |
| Oversea1 | 0.441*** | | -0.080 | |
| | (0.159) | | (0.148) | |
| Oversea2 | | 1.360** | | 0.430 |
| | | (0.592) | | (0.634) |
| Oversea1*SOE | 1.890*** | | | |
| | (0.245) | | | |
| Oversea2*SOE | | 6.759*** | | |
| | | (0.975) | | |
| Oversea1* Scale | | | 2.416*** | |
| | | | (0.239) | |
| Oversea2* Scale | | | | 6.044*** |
| | | | | (0.909) |
| Size | 2.251*** | 2.280*** | 2.103*** | 2.211*** |
| | (0.060) | (0.061) | (0.062) | (0.062) |
| ListAge | -0.697*** | -0.701*** | -0.667*** | -0.674*** |
| | (0.095) | (0.094) | (0.094) | (0.094) |
| Lev | -2.584*** | -2.583*** | -2.599*** | -2.541*** |
| | (0.347) | (0.349) | (0.348) | (0.350) |
| Roe | 2.979*** | 2.912*** | 2.974*** | 2.861*** |
| | (0.418) | (0.418) | (0.421) | (0.421) |
| Growth | -0.271** | -0.271** | -0.301** | -0.311*** |
| | (0.118) | (0.118) | (0.118) | (0.119) |
| Board | 1.255*** | 1.438*** | 1.407*** | 1.525*** |
| | (0.309) | (0.307) | (0.307) | (0.306) |
| Indep | 0.068*** | 0.071*** | 0.071*** | 0.073*** |
| | (0.011) | (0.011) | (0.011) | (0.011) |
| Dual | -0.056 | -0.048 | -0.109 | -0.086 |
| | (0.134) | (0.135) | (0.134) | (0.135) |
| Top1 | -0.005 | -0.006* | -0.003 | -0.005 |
| | (0.004) | (0.004) | (0.004) | (0.004) |
| _cons | -26.520*** | -27.579*** | -23.725*** | -26.362*** |
| | (1.424) | (1.425) | (1.413) | (1.421) |
| Year FE | Yes | Yes | Yes | Yes |
| Industry FE | Yes | Yes | Yes | Yes |
| adj. $R^2$ | 0.648 | 0.648 | 0.646 | 0.648 |
| F | 200.735 | 197.177 | 196.269 | 197.177 |
| N | 12949 | 12949 | 12949 | 12949 |

Note: In parentheses are robust standard errors, * * *, * * and * are significance levels of 1%, 5% and 10%, respectively.

**6.2.3 Analysis of regional heterogeneity.** Regional disparities in development levels may lead to heterogeneous effects of overseas background executives on corporate ESG performance. To investigate this, we categorize the sample into Eastern region and Central-Western region groups based on national policies and provincial economic development levels, re-estimating Model (1). Results are presented in Columns (1) and (2) of **Table 13**. we find that the interaction terms *Oversea1\*East* and *Oversea2\*East* yield coefficients of 1.286 and 3.659, respectively, both statistically significant at the 1% level. This indicates that overseas background executives exert a stronger ESG-enhancing effect in the economically advanced Eastern region compared to the less-developed Central and Western regions.

Maybe because the Eastern region's more mature market mechanisms and lower governmental intervention create an environment where firms face heightened demands for sustainability from governments, society, and investors. To gain a competitive edge, overseas background executives in this region are incentivized to prioritize ESG improvements, leveraging their expertise to align corporate strategies with market expectations.

**6.2.4 Analysis of industry heterogeneity.** The motivations and objectives for adopting ESG strategies differ significantly between firms in heavily polluting industries and those in other sectors. To explore this heterogeneity, we divide the sample into two groups: firms in heavily polluting industries and those in non-polluting industries, re-estimating Model (1). Results are presented in Columns (3) and (4) of **Table 13**. We find that the interaction terms *Oversea1\*Pollute* and *Oversea2\*Pollute* yield coefficients of 1.627 and 4.765, respectively, are statistically significant at the 1% level. This indicates that overseas background executives exert a more pronounced ESG-enhancing effect in heavily polluting industries compared to other sectors.

This may be because, in the reality of advocating green development, firms in heavily polluting industries face greater cost pressures and environmental regulatory pressures. To comply with national and local environmental regulations, protect the environment, and promote sustainable development, these firms have stronger incentives to improve ESG performance. Overseas-experienced executives, leveraging their expertise in global sustainability practices, play a critical role in driving such improvements, aligning corporate operations with stricter environmental and social governance standards.

## 7. Conclusions

As China's comprehensive national power has significantly strengthened, a growing number of overseas-educated professionals have returned to join domestic firms as executives, contributing to corporate sustainable development. Concurrently, ESG (Environmental, Social, and Governance) practices—an investment philosophy aligned with sustainability—have gained unprecedented attention across society. Using data from Chinese A-share listed companies (2008–2022), this paper first empirically examines the impact of overseas background executives on corporate ESG performance. then we analyze how internal and external executive compensation gaps moderate this relationship. Additionally, the study explores heterogeneity in these effects across firm ownership, size, region, and industry. The primary findings are as follows:

First, overseas background executives significantly enhance ESG performance, and a higher proportion of such executives within the leadership team correlates with stronger ESG outcomes. These conclusions withstand rigorous robustness checks.

Second, both internal and external executive pay gaps positively moderate the relationship between overseas background executives and ESG performance. Under China's policy mandating public disclosure of executive compensation, overseas background executives prioritize long-term corporate interests, leading to increased ESG investments.

Third, heterogeneity exists in the ESG-enhancing effects of overseas-experienced executives: The effects are stronger in state-owned enterprises, larger firms, eastern regions, and heavily polluting industries.

### 7.1 Theoretical implications

This study enriches the existing body of knowledge in multiple dimensions.

**Table 13. Results of regional and industry heterogeneity analysis.**

| | Heterogeneity of region | | Heterogeneity of industry | |
| --- | --- | --- | --- | --- |
| | (1) | (2) | (3) | (4) |
| *Variables* | *ESG* | *ESG* | *ESG* | *ESG* |
| Oversea1 | 0.264 | | 0.852*** | |
| | (0.234) | | (0.143) | |
| Oversea2 | | 0.985 | | 2.967*** |
| | | (1.148) | | (0.517) |
| Oversea1* East | 1.286*** | | | |
| | (0.268) | | | |
| Oversea2* East | | 3.659*** | | |
| | | (1.227) | | |
| Oversea1* Pollute | | | 1.627*** | |
| | | | (0.341) | |
| Oversea2* Pollute | | | | 4.765*** |
| | | | | (1.365) |
| Size | 2.281*** | 2.311*** | 2.287*** | 2.316*** |
| | (0.060) | (0.061) | (0.060) | (0.061) |
| ListAge | -0.566*** | -0.590*** | -0.626*** | -0.625*** |
| | (0.096) | (0.095) | (0.095) | (0.094) |
| Lev | -2.407*** | -2.420*** | -2.445*** | -2.469*** |
| | (0.349) | (0.351) | (0.348) | (0.350) |
| Roe | 2.941*** | 2.839*** | 2.784*** | 2.753*** |
| | (0.420) | (0.420) | (0.422) | (0.421) |
| Growth | -0.299** | -0.301** | -0.315*** | -0.316*** |
| | (0.119) | (0.119) | (0.118) | (0.119) |
| Board | 1.398*** | 1.494*** | 1.374*** | 1.547*** |
| | (0.309) | (0.307) | (0.307) | (0.305) |
| Indep | 0.070*** | 0.071*** | 0.071*** | 0.072*** |
| | (0.011) | (0.011) | (0.011) | (0.011) |
| Dual | -0.152 | -0.119 | -0.155 | -0.118 |
| | (0.135) | (0.135) | (0.135) | (0.135) |
| Top1 | -0.003 | -0.004 | -0.003 | -0.004 |
| | (0.004) | (0.004) | (0.004) | (0.004) |
| _cons | -28.077*** | -28.814*** | -28.003*** | -28.971*** |
| | (1.413) | (1.421) | (1.419) | (1.421) |
| Year FE | Yes | Yes | Yes | Yes |
| Industry FE | Yes | Yes | Yes | Yes |
| adj. $R^2$ | 0.646 | 0.646 | 0.646 | 0.646 |
| F | 195.758 | 193.719 | 196.269 | 193.893 |
| N | 12949 | 12949 | 12949 | 12949 |

Note: In parentheses are robust standard errors, $***$, $**$ and $*$ are significance levels of 1%, 5% and 10%, respectively.

First, by systematically examining the appointment of overseas background executives in corporate settings, it enhances scholarly attention to the literature on executives with international backgrounds, particularly in the context of Chinese listed companies.

Second, through empirical evidence, we demonstrate the significant impact of overseas background executives on corporate ESG performance. This contribution advances the literature on ESG outcomes and provides novel insights into Upper Echelons Theory, illustrating how executives' experiences and values shape organizational decisions and sustainability practices.

Third, the study delves into the moderating roles of internal and external executive pay gaps in the relationship between overseas background executives and ESG performance—a topic rarely explored in prior research. This deepens the understanding of Tournament Theory by highlighting how incentive structures interact with leadership characteristics to drive ESG-related outcomes.

### 7.2 Practical implications

The findings of this study offer actionable insights for policymakers and corporate leaders.

First, our government should vigorously enhance ESG regulatory systems, actively promote ESG development principles, and prioritize corporate ESG performance to provide clear guidance for long-term sustainable development. At the same time, our government should also encourage enterprises across industries to recruit high-caliber overseas talent, thereby optimizing talent recruitment and team building to elevate corporate ESG performance.

Second, enterprises should establish long-term sustainability goals, deepen their understanding of ESG principles, integrate ESG considerations into strategic planning and decision-making systems, and emphasize balanced performance across economic, social, and environmental dimensions.

Third, enterprises should further refine their corporate governance structures, talent recruitment policies, and executive compensation frameworks. They must recognize and leverage the strategic value of overseas background executives in ESG-related decision-making, ensuring their expertise drives sustainable long-term corporate growth.

### 7.3 Limitations and future research

Despite the valuable insights provided by this study, there are still some limitations. First, while this study focuses on executives' overseas backgrounds, it does not provide a granular analysis of these backgrounds—such as the countries of origin, duration of overseas exposure, or hierarchical authority of the executives. Additionally, other leadership characteristics (e.g., academic training and financial expertise) may shape decision-making and thereby influence corporate ESG performance. Future research should conduct detailed examinations of these dimensions to deepen understanding of how overseas experiences interact with other attributes to drive sustainability outcomes.

Second, while this study examines corporate ESG performance through the lens of executives with overseas backgrounds, numerous other internal factors (e.g., board diversity, innovation incentives) and external determinants (e.g., regulatory pressures, stakeholder activism) remain underexplored. Future studies could broaden the scope by systematically investigating these additional dimensions to map the drivers of ESG outcomes.

Third, while this study examines the impact of overseas-experienced executives on corporate ESG performance, it lacks an in-depth analysis of the underlying mechanisms. Future research should prioritize exploring the pathways through which such executives influence ESG outcomes, thereby refining and expanding the current body of knowledge.

### Acknowledgments

The authors sincerely thank the editors and the anonymous reviewers for their helpful comments and suggestions. All remaining errors are our own.

### Author contributions

**Conceptualization:** Jing Liu.

**Data curation:** Haohao Wei.

**Formal analysis:** Baoping Liu, Haohao Wei.

**Funding acquisition:** Baoping Liu.

**Methodology:** Baoping Liu, Jing Liu.

**Project administration:** Haohao Wei.

**Software:** Haohao Wei.

**Writing – original draft:** Baoping Liu.

**Writing – review & editing:** Baoping Liu.

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
