## [Decision Letter · Decision Letter 0]

23 Jan 2025

PONE-D-24-50043Overseas background executives and enterprise ESG performance--the moderating effect of the executive pay gapPLOS ONE

Dear Dr. Liu,

Thank you for submitting your manuscript to PLOS ONE. After careful consideration, we feel that it has merit but does not fully meet PLOS ONE’s publication criteria as it currently stands. Therefore, we invite you to submit a revised version of the manuscript that addresses the points raised during the review process.

We look forward to receiving your revised manuscript.

Kind regards,

Wafa Ghardallou

Academic Editor

PLOS ONE

Journal Requirements:

This work is supported by the Major Program Grants for Applied Research from the Education Department of Henan Province (2023-YYZD-04), China.

4. Your abstract cannot contain citations. Please only include citations in the body text of the manuscript, and ensure that they remain in ascending numerical order on first mention.

Reviewers' comments:

Reviewer's Responses to Questions

**Comments to the Author**

1. Is the manuscript technically sound, and do the data support the conclusions?

Reviewer #1: Yes

Reviewer #2: Yes

2. Has the statistical analysis been performed appropriately and rigorously? 

Reviewer #1: Yes

Reviewer #2: No

3. Have the authors made all data underlying the findings in their manuscript fully available?

Reviewer #1: No

Reviewer #2: Yes

4. Is the manuscript presented in an intelligible fashion and written in standard English?

Reviewer #1: No

Reviewer #2: No

5. Review Comments to the Author

Reviewer #1: This study explores the impact of executives with overseas backgrounds on corporate ESG performance. While the study provides a solid theoretical foundation, a significant concern is the lack of innovation in both the research design and methodology. The detailed comments are listed below.

1. The primary concern lies in the lack of novelty in this study. All the variables have already been explored in existing research, and the study fails to demonstrate sufficient innovation in terms of either data collection or methodology. The authors addressed that “Despite these insights, detailed research on the relationship between overseas background executives and corporate ESG performance remains sparse both domestically and internationally. Therefore, there is ample scope for further investigation into the impact of overseas background executives on corporate ESG performance.” However, a recent study from Liu et al. (2024) have examined the impact of the Chinese CEOs with foreign experience on the firm's ESG performance. Please justify the differences. Additionally, many studies have examined the connection between Executive Compensation and ESG Performance, as detailed in the reference section.

The authors need to identify and articulate what sets their study apart from previous research. I recommend the authors can try to explore additional dimensions of CEO overseas experience (e.g., duration, type of overseas exposure, or specific skills gained) instead of using a binary dummy variable. This could provide deeper insights into its impact on ESG performance. Besides, the authors may consider exploring the mechanisms through which overseas executives influence ESG performance.

2. While the significance of overseas executives is highlighted, it would help to specify why this focus is particularly relevant to China's current stage of economic development.

3. While stakeholder theory, top echelon theory, and imprinting theory are presented, their integration feels somewhat fragmented. The narrative jumps between theories without clearly demonstrating how they build upon one another to explain the observed phenomena.

4. When the authors consider the impact of specific years, they select the 2015 China financial crisis and the 2020 COVID-19 pandemic as two key events for the Year Impact Test, focusing on their relevance to the high-quality development of enterprises. However, why is the 2008 global financial crisis not included in this analysis?

5. The study found that the influence of overseas executives on ESG performance is more pronounced in state- owned enterprises (SOEs), large-scale firms, firms in the eastern region, and those in high-pollution industries. However, the analysis section lacks further explanation and provides only a general statistical description.

6. It is recommended that the study's limitations be included at the end of the study.

References

Liu, Y., Zhang, F., & Zhang, H. (2024). CEO foreign experience and corporate environmental, social, and governance (ESG) performance. Business Strategy and the Environment, 33(4), 3331–3355. https://doi.org/10.1002/bse.3647

COHEN, S., KADACH, I., ORMAZABAL, G. and REICHELSTEIN, S. (2023), Executive Compensation Tied to ESG Performance: International Evidence. Journal of Accounting Research, 61: 805-853. https://doi.org/10.1111/1475-679X.12481

Meng, T., Lu, D., Yu, D. et al. Is executive compensation aligned with the company’s ESG objectives? Evidence from Chinese listed companies based on the PSM-DID approach. Humanit Soc Sci Commun 11, 1560 (2024). https://doi.org/10.1057/s41599-024-04094-y

Zhu, C., Liu, X., Chen, D., & Yue, Y. (2024). Executive compensation and corporate sustainability: Evidence from ESG ratings. Heliyon, 10(12), e32943. https://doi.org/10.1016/j.heliyon.2024.e32943

Reviewer #2: I have carefully reviewed the manuscript which investigates the impact of executives with overseas backgrounds on corporate ESG performance, utilizing a two-way fixed-effects model based on data from Chinese A-share listed companies in Shanghai and Shenzhen spanning from 2008 to 2022. The subject is very important for the literature. The paper is informative. However, due to the following problems, my decision is a major revision. Please see my comments and suggestions below.

Comment 1. The first paragraph of the introduction should be rewritten, please focus on the research topic.

Comment 2. The Introduction needs to be rearranged, and more work is needed to strengthen the theoretical basis. Additionally, how this study bridges the research gaps should be elucidated. The following paper can be a good example to help you improve your paper (Does climate change exposure impact on corporate finance and energy performance? Unraveling the moderating role of CEOs’ green experience. Journal of Cleaner Production, 461, 142653).

Comment 3. Please elaborate on the specific content of this study in the introduction. Because the introduction has not yet mentioned how the research was conducted to arrive at the research contribution of this paper. The following paper can be a good example to help you improve your paper (Does firm‐level exposure to climate change influence inward foreign direct investment? Revealing the moderating role of ESG performance. Corporate Social Responsibility and Environmental Management, 31(6), 6167-6183).

Comment 4. Please state the remain structure of this study in the introduction.

Comment 5. The manuscript falls short in terms of theoretical development. What theories can be used as the theoretical basis for your research topic?

Comment 6. To better clarify the research hypotheses carried out I suggest elaborating a figure of theoretical framework in the section 2.

Comment 7. The measurement of all variables should be supported by references.

Comment 8. In the sections 4 and 6, the authors seem to "only report the results". I suggest that authors use several representative studies (2 or 3) in this area to interpret and enrich the results.

Comment 9. The authors should present the theoretical and practical implications, limitations and future research in the section conclusion.

Comment 10. The language of this paper is very bad and needs help from native speakers.

Good luck with your work!

6. PLOS authors have the option to publish the peer review history of their article (what does this mean? ). If published, this will include your full peer review and any attached files.

**Do you want your identity to be public for this peer review?** For information about this choice, including consent withdrawal, please see our Privacy Policy .

Reviewer #1: No

Reviewer #2: No

---

## [Editor Report · Decision Letter 1]

29 Apr 2025

Overseas background executives and enterprise ESG performance - the moderating effect of the executive pay gap

PONE-D-24-50043R1

Dear Dr. Liu,

We’re pleased to inform you that your manuscript has been judged scientifically suitable for publication and will be formally accepted for publication once it meets all outstanding technical requirements.

Kind regards,

Wafa Ghardallou

Academic Editor

PLOS ONE
---

## [Editor Report · Acceptance letter]

PONE-D-24-50043R1

PLOS ONE

Dear Dr. Liu,

I'm pleased to inform you that your manuscript has been deemed suitable for publication in PLOS ONE. Congratulations! Your manuscript is now being handed over to our production team.

Kind regards,

on behalf of

Dr. Wafa Ghardallou

Academic Editor

PLOS ONE